# Cerebral Cavernous Malformation Pathogenesis: Investigating Lesion Formation and Progression with Animal Models

**DOI:** 10.3390/ijms23095000

**Published:** 2022-04-30

**Authors:** Chelsea M. Phillips, Svetlana M. Stamatovic, Richard F. Keep, Anuska V. Andjelkovic

**Affiliations:** 1Neuroscience Graduate Program, University of Michigan Medical School, Ann Arbor, MI 48109, USA; chelsphi@med.umich.edu; 2Department of Pathology, University of Michigan Medical School, Ann Arbor, MI 48109, USA; sstamato@med.umich.edu; 3Department of Neurosurgery, University of Michigan Medical School, Ann Arbor, MI 48109, USA; rkeep@med.umich.edu; 4Molecular and Integrative Physiology, University of Michigan Medical School, Ann Arbor, MI 48109, USA

**Keywords:** cerebrovascular malformation, KRIT1, CCM2, PDCD10, hemorrhagic lesions, CCM animal models, RhoA kinase inhibition

## Abstract

Cerebral cavernous malformation (CCM) is a cerebromicrovascular disease that affects up to 0.5% of the population. Vessel dilation, decreased endothelial cell–cell contact, and loss of junctional complexes lead to loss of brain endothelial barrier integrity and hemorrhagic lesion formation. Leakage of hemorrhagic lesions results in patient symptoms and complications, including seizures, epilepsy, focal headaches, and hemorrhagic stroke. CCMs are classified as sporadic (sCCM) or familial (fCCM), associated with loss-of-function mutations in *KRIT1/CCM1*, *CCM2*, and *PDCD10/CCM3.* Identifying the CCM proteins has thrust the field forward by (1) revealing cellular processes and signaling pathways underlying fCCM pathogenesis, and (2) facilitating the development of animal models to study CCM protein function. CCM animal models range from various murine models to zebrafish models, with each model providing unique insights into CCM lesion development and progression. Additionally, these animal models serve as preclinical models to study therapeutic options for CCM treatment. This review briefly summarizes CCM disease pathology and the molecular functions of the CCM proteins, followed by an in-depth discussion of animal models used to study CCM pathogenesis and developing therapeutics.

## 1. Introduction

Cerebral cavernous malformation (CCM), a disease affecting the cerebrovascular capillary beds, affects up to 0.5% of the population [1]. Hallmarks of CCM include vessel dilation, decreased contact between endothelial cells lining the vessels, loss of junctional complex proteins, and a compromised brain endothelial barrier resulting in blood-filled caverns [2]. Ultimately, loss of brain endothelial barrier integrity leads to patient symptoms and complications, including seizures, epilepsy, focal headaches, and hemorrhagic stroke [1]. Current CCM treatments include surgical resection of the lesion, gamma knife surgery, or symptom alleviation, highlighting the need for novel therapeutics [1,3,4,5]. CCMs are either sporadic (sCCM) or familial/inherited (fCCM), which are associated with loss-of-function mutations in *KRIT1/CCM1*, *CCM2*, or *PDCD10/CCM3* [6,7,8,9,10]. Mutations are inherited in an autosomal dominant manner, with mutation of the remaining functional allele, a ‘second hit’, leading to disease onset [11,12,13,14]. Out of the three types of fCCM, fCCM3 is the most aggressive, with an earlier age of disease onset in fCCM3 patients and increased disease severity [15,16,17,18]. Discovering the CCM proteins has allowed for the elucidation of their endogenous functions and signaling pathways and the generation of animal models. This review provides an overview of CCM disease pathology and CCM protein function, along with a detailed discussion of murine and zebrafish CCM models and promising therapeutics to treat CCM.

## 2. CCM Clinical Presentation

Characterized by their “mulberry-like” presentation, CCM lesions consist of vascular sinusoids lacking intervening brain tissue and microvasculature components, including astrocytic endfeet, pericytes, and smooth muscle cells [1,2,19,20]. Reactive astrocytes, macrophages containing hemosiderin, and glial cells are located peripheral to the lesion [2,19,20]. Histological analyses have demonstrated that lesional endothelial cells lack tight junction proteins that mediate cell–cell adhesion, resulting in loss of cell–cell contact and basal lamina exposure [2]. Together, loss of the microvascular structural support and tight junction proteins result in brain endothelial barrier permeability [2].

CCM diagnosis is conducted via T_2_-weighted magnetic resonance imaging (MRI): large CCMs present as a ring of decreased signal intensity enclosing an area of variable signal intensity and small CCMs appear as low signal intensity puncta [19,21,22]. While 11–44% of CCM patients remain asymptomatic, 23–50% of CCM patients clinically present with seizures, 6–52% with headaches, 20–45% with focal neurological deficits, and 9–56% with hemorrhage [1,22]. Up to 80% of CCM lesions are supratentorial, localized to the frontal or temporal lobes; however, infratentorial lesions are associated with a worsened prognosis [1,22,23,24]. Previous hemorrhage is also associated with a worsened prognosis, but lesion size and number are not [23]. Although certain studies identify a sex bias with an increased number of female CCM patients, most studies have not investigated sex as a biological variable [1,20,23,25,26]. The presence of multiple lesions is common in fCCM patients (>50% of cases) and rare in sCCM patients (12–20%) [1]. Furthermore, fCCM3 has an increased disease severity compared to fCCM1 and fCCM2, with an earlier age of onset, higher initial incidence of cerebral hemorrhage, an association with dural-based lesions, and increased seizure incidence [15,16,17,18].

## 3. CCM Proteins: Structure, Expression Patterns, and Molecular Functions

Loss-of-function mutations in *KRIT1/CCM1*, *CCM2*, and *PDCD10/CCM3* provide molecular substrates to understand CCM disease pathogenesis, emphasizing the importance in understanding their endogenous functions. This section summarizes the molecular functions of each of the CCM proteins, along with the function of the CCM signaling complex (CSC). 

### 3.1. KRIT1/CCM1

Discovered via a two-hybrid screen for Krev-1/rap1a interactors, Krev interaction trapped protein 1 (KRIT1) consists of an N-terminal domain with four ankyrin repeats and a C-terminal domain required for Krev-1 interactions [27]. During murine development, *Krit1* transcript is expressed across tissue types, including neuronal cell layers, the heart, and large vessels; however, *Krit1* is not expressed in meningeal or parenchymal cerebral vessels [28]. While neuronal expression of *Krit1* mRNA is detected in adulthood, heart and large vessel expression declines across murine development [28]. KRIT1 is expressed in the vascular endothelium of various human organs, with high expression in blood–organ barriers but not fenestrated capillaries [29]. Within cerebral tissue, KRIT1 is expressed in endothelial cells, astrocytic endfeet, and pyramidal neurons [29]. At the subcellular level, KRIT1 localizes to the nucleus and cytoplasm, with localization regulated by protein kinase Cɑ (PKCɑ) [30,31]. KRIT1 interacts with Rap1 via the FERM domain of KRIT1, an interaction required for Rap1-mediated stabilization of cell–cell junctions [30]. KRIT1 depletion increases RhoA kinase and Rho kinase (ROCK) activity, resulting in the formation of actin stress fibers [30,32,33]. Endothelial loss of KRIT1 also results in upregulation of MEKK3/KLF2/4 signaling in vivo [34]. Additionally, KRIT1 has a well-defined role in development: *Krit1*^−/−^ mice die mid-gestation from vascular defects, and *san*, the zebrafish homolog of *KRIT1*, is required for thickening of the myocardial wall [35,36].

### 3.2. CCM2

The *CCM2* loci was mapped to 7p13–15, ultimately leading to the identification of malcavernin, or CCM2, a protein with a phosphotyrosine binding (PTB) domain [9,37]. *Ccm2* expression patterns during murine development mirror *Ccm1*, with transcript expression in neuronal cell layers, the heart, and large vessels; however, *Ccm2* is also developmentally expressed in meningeal cerebral vessels [28]. Likewise, human CCM2 expression patterns are comparable to that of KRIT1, as CCM2 is localized to endothelial cells, astrocytes and their endfeet, and pyramidal neurons in the cerebral cortex [38]. Subcellularly, CCM2 primarily localizes to the cytoplasm but also to the nucleus, despite lacking a nuclear localization signal and nuclear export signal [39]. Subcellular localization of CCM2 is likely regulated through its interaction with KRIT1 via PTB binding to KRIT1 NPXY motifs [39]. Depletion of CCM2 results in a similar phenotype as KRIT1 depletion, as loss of CCM2 increases GTP-bound RhoA and leads to actin stress fiber formation [40]. Similarly, CCM2 is essential in development. *Ccm2*^−/−^ mice die mid-gestation from failed angiogenesis, and the zebrafish homolog of *CCM2* is required for myocardial wall thickening [36,40,41,42]. CCM2 also interacts with CCM3, forming a complex with serine/threonine kinase 25 (STK25) [43].

### 3.3. PDCD10/CCM3

Programmed cell death 10 (PDCD10), originally named TF-1 apoptosis related gene-15 (TRAF15), was identified during a screen for apoptosis-related genes in growth serum-deprived TF-1 cells, with its expression protecting against natural cell death [44]. PDCD10 is a two-domain protein, consisting of an N-terminal dimerization domain and a C-terminal focal adhesion targeting (FAT)-homology domain [45]. Initial characterization of expression patterns demonstrated that PDCD10 is expressed in fetal liver, kidney, lung, and spleen [44]. Similar to *Ccm2*, *Pdcd10* transcript expression is detected in neuronal cell layers, the heart, large vessels, and meningeal vessels during murine development [28]. Within human cerebral tissue, the expression pattern of PDCD10 follows that of KRIT1 and CCM2: PDCD10 is expressed in the arterial endothelium and glial cells, with little to no expression in the venous endothelium [46]. At a subcellular level, PDCD10 localizes to the Golgi apparatus via interactions with the Golgi protein GM130; PDCD10 depletion results in Golgi apparatus disassembly [47]. PDCD10 interacting partners include CCM2 via the PDCD10 FAT-homology domain, STK25, STK26/MST4, and Fas-associated phosphatase-1 (FAP-1) [43,45,48]. STK25 and FAP-1 phosphorylate and dephosphorylate PDCD10, respectively [43]. PDCD10 stabilizes STKs, a highly conserved interaction, negatively regulating Rho via moesin and promoting endothelial barrier integrity [47,49,50]. The subsequent increase in ROCK activity following loss of PDCD10 has also been observed in vivo and in human CCM3 lesions [51]. Additionally, PDCD10-STK26/MST4 interactions promote cell growth and transformation through ERK signaling [48]. Similar to KRIT1 and CCM2, PDCD10 is crucial for vascular development. Loss of PDCD10 decreases VEGFR2 stability and leads to failed angiogenesis [52,53].

### 3.4. CCM Signaling Complex (CSC)

Although the interaction between KRIT1 and CCM2 had been previously documented, proteomic analysis identified the CSC: the ternary complex formed by KRIT1, CCM2, and PDCD10 [54,55]. Within the CSC, CCM2 simultaneously interacts with KRIT1 and PDCD10; *CCM2* mutations that disrupt KRIT1–CCM2 interactions abolish CSC formation [43,56]. With a wide range of interacting partners, the CSC is involved in various cellular processes, including actin cytoskeleton organization, cell adhesion, migration, proliferation, and apoptosis [32,55,57,58,59]. While belonging to the CSC, PDCD10 primarily interacts with the striatin-interacting phosphatase and kinase (STRIPAK) complex, an enzymatic network with functions including endocytosis, Golgi development, and calcium sensing [60,61]. The interaction between PDCD10 and the STRIPAK complex may explain the increased disease severity in fCCM3 patients compared to fCCM1 and fCCM2 patients [15,16,17,18].

## 4. CCM Pathogenesis

### 4.1. CCM Lesion Formation

#### 4.1.1. Familial CCM

Pathogenesis of fCCM is thought to follow the Knudson two-hit hypothesis, where patients are born with one null allele and one functional allele [11,12,13,14,62]. Although currently unknown, the second hit results in a loss-of-function mutation in the remaining allele, leading to CCM lesion formation [11,12,13,14]. Identification of somatic mutations in addition to germline mutations within human CCM lesions validates the two-hit mechanism of pathogenesis [12,13,14]. Following loss of the remaining functional allele, CCM-null endothelial cells begin expressing mesenchymal markers and undergo clonal expansion to generate cavernomas [63,64]. As cavernomas grow, they recruit wild-type endothelial cells, yielding the observed endothelial cell mosaicism in CCM lesions [62,63,64,65].

#### 4.1.2. Sporadic CCM

Because of the described two-hit mechanism of fCCM pathogenesis, sCCM pathogenesis was hypothesized to follow the same two-hit mechanism [66]. This hypothesis was further confirmed through the identification of somatic mutations in *CCM* genes in sCCM patients [66,67,68]. Recent studies, however, refute this hypothesis. Only 10% of sCCM lesions contain mutations within *CCM* genes, and *CCM* transcript expression is not dysregulated [69,70,71]. These studies have identified somatic *PIK3CA* and *MAP3K3* mutations in sCCM lesions and highlight other molecular substrates in sCCM pathogenesis, such as PI3K/Akt/mTOR signaling, epilepsy-associated genes, neuroinflammation, metabolism of reactive oxygen species, remodeling of the extracellular matrix, and deterioration of cell junctions [69,70,71,72].

### 4.2. CCM Lesion Progression

The main hypothesis surrounding CCM lesion progression is that lesion leakage causes patient symptoms and complications, supported by the finding that vascular permeability in the brain and lesions are viable biomarkers for CCM disease progression [73,74,75]. Studies conducted with human CCM patients and murine CCM models highlight inflammation as a contributor to CCM progression, with pro-inflammatory cytokine expression and immune cell infiltration of lesions correlating with lesion progression and patient symptoms [76,77,78,79,80]. Other implicated cell processes contributing to disease progression include increased RhoA kinase activity and PI3K/Akt/mTOR signaling [81,82,83]. Ultimately, understanding the cellular processes driving lesion progression is essential to develop therapeutics for CCM lesion stabilization.

## 5. Animal Models

The discovery of mutations in *KRIT1/CCM1*, *CCM2*, and *PDCD10/CCM3* has facilitated in vivo fCCM modeling, resulting in the development of various animal models. To date, there are no animal models of sCCM. This section discusses the various murine and zebrafish animal models used to model fCCM pathogenesis, along with the insights each model provides.

### 5.1. Murine Models

#### 5.1.1. Global Loss of CCM

Globally knocking out CCM proteins results in embryonic lethality due to failed angiogenesis [35,40,41,42,52,53]. While these animal models are not viable for studying CCM pathogenesis, they have proven useful in elucidating the developmental roles of CCM proteins. KRIT1 is required for vascular development, with loss of KRIT1 resulting in increased arterial dilatation, narrowing of branchial arch arteries and rostral dorsal aorta, and decreased expression of arterial markers and Notch genes [35]. Global loss of CCM2 reveals CCM2 expression is crucial for developmental processes such as angiogenesis, lumen formation of the first branchial arch artery, and heart development [40,42,84]. Similar to KRIT1 and CCM2, PDCD10 is involved in angiogenesis; however, PDCD10 loss also results in decreased VEGFR2 expression and activity [52]. Interestingly, while each of the CCM proteins are essential for proper angiogenesis, global loss of PDCD10 results in earlier growth arrest and death compared to global loss of KRIT1 and CCM2 [53].

#### 5.1.2. Sensitized CCM Models

Due to the autosomal dominant inheritance pattern of fCCM, mouse models heterozygous for one of the CCM proteins were developed. Mice heterozygous for CCM proteins develop CCM lesions, particularly *Ccm3*^+/*−*^ mice; however, penetrance is low [51,85]. To increase penetrance, sensitized CCM models were generated via crossing mice heterozygous for one of the CCM proteins with a genetically unstable mouse line, such as *Trp53^−/−^* or *Msh2^−/−^* mice [86]. Following the proposed two-hit mechanism, increased genetic instability results in loss of the remaining functional *Ccm* allele [86]. This strategy has proven effective, as *Ccm1^+/−^Msh2^−/−^*, *Ccm1^+/−^Trp53^−/−^,* and *Ccm2^+/−^Trp53^−/−^* mice develop CCM lesions by 4 to 5 months of age [86,87,88]. Lesion development in *Ccm3^+/−^Trp53^−/−^* mice has been observed as early as 5 weeks of age [89]. MRI and histological analyses of the sensitized CCM model demonstrate that individual CCM lesions in a given animal range in size and stage of lesion progression [86,87,88,89]. Importantly, these sensitized CCM lesions share characteristics with human CCM lesions, including capillary dilation, iron deposition, immune cell infiltrates, endothelial cell proliferation, and increased ROCK activity [86,87,88]. Interestingly, sensitized CCM3 mice have an increased lesion burden compared to sensitized CCM1 and CCM2 mice, mimicking the increased severity of fCCM3 [51]. Although CCM lesions in the sensitized model faithfully recapitulate defining characteristics of human CCM lesions, a potential confounding variable is the propensity of mice with genetically unstable backgrounds to develop tumors: *Msh2*^-/-^ mice develop lymphomas around two months and *Trp53*^−/−^ mice develop neoplasms by six months [90,91,92]. Tumor development in the sensitized CCM model is rare but has been documented for *Ccm1^+/−^* and *Ccm2^+/−^* mice with a *Trp53*^−/−^ background [88]. Using the sensitized CCM model proves challenging due to the difficult breeding scheme and a low yield of animals with the desired genotype [93]. Ultimately, the sensitized CCM model relies on a genetically unstable background for loss of the remaining functional *Ccm* allele, resulting in murine CCM lesions that mimic human CCM lesions.

#### 5.1.3. Conditional CCM Models

Murine models with conditional loss of CCM expression has elucidated cell types requiring CCM protein expression for proper development. Endothelial cell-specific loss of CCM2 is embryonically lethal, but neuronal, smooth muscle, and neuroepithelial loss of CCM2 produces viable mutants [40,41,42,52,53]. Likewise, endothelial loss of CCM3 is embryonically lethal, yet mice lacking smooth muscle CCM3 survive [52,53]. Investigating neuronal- and glial-specific CCM3 loss using the same *Nestin*-Cre,*Ccm3*-flox mice has generated conflicting results: one study found *Nestin*-Cre,*Ccm3*-flox mice are viable, while another demonstrated *Nestin*-Cre,*Ccm3*-flox mice have enlarged brains and only survive to post-natal day 3 (P3) [52,94]. Astrocytic-, neuronal-, and mural-specific CCM3 deletion with *Gfap*-Cre, *Emx1*-Cre, and *Sm22**ɑ*-Cre lines, respectively, results in CCM lesion development [94,95]. *Gfap*-Cre,*Ccm3*-flox mice develop brain and spinal cord lesions as early as 3 weeks and *Emx1*-Cre,*Ccm3*-flox mice develop forebrain lesions at 7 months [94]. These findings are unique, as CCM has been thought to be an endothelial cell autonomous disease [62]. Together, conditional CCM models demonstrate the necessity of endothelial CCM protein expression for development.

#### 5.1.4. Inducible CCM Models

With each previously described CCM model providing its own set of caveats, tamoxifen-inducible Cre recombinase (CreERT2) technology was harnessed to generate inducible CCM knockout mice. In this CCM model, *Ccm*-floxed mice are crossed with CreERT2 mouse lines, such as *Pdgfb*-CreERT2 or *Cdh5*(PAC)-CreERT2 mice for endothelial-specific CCM loss and *Slco1c1*(BAC)-CreERT2 or *Mfsd2a*CreER^T2^ mice for brain endothelial-specific CCM loss [53,96,97,98]. Recently, loss of CCM3 in endothelial progenitor cells has been achieved using *Procr*^CreERT2-IRES-tdTomato/+^ mice [99]. To induce CCM lesion formation, tamoxifen-inducible Cre recombinase expression and the resulting loss of CCM expression must occur between P1 and P4 [53,96]. Tamoxifen (TMX) treatment at P8, P15, and P21 does not result in CCM lesion formation in *Ccm2*^+/−^;*Cdh5*(PAC)-CreERT2 mice [96]. Inducible CCM1 and CCM2 models develop CCM lesions restricted to the hindbrain and retina around P6, with a 17-day survival median [34,96]. To combat the early mortality rate of the inducible models, chronic CCM models are generated through the administration of single low dose of TMX post-birth. These chronic models develop lesions with chronic hemorrhage and immune cell infiltrates throughout the brain and survive to P90 [63,100]. For CCM2 models, brain endothelial-specific loss of CCM2 results in a chronic model with a 6-month survival median, while general endothelial cell CCM2 deletion yields an acute model [96,101]. Inducible CCM models develop CCM lesions sharing similarities with human CCM lesions; for example, endothelial cell-specific CCM3 deletion at P1 results in CCM lesions with hemosiderin deposits (a marker of cerebral hemorrhage), lack of pericytes and astrocytic foot processes, and mononuclear inflammatory cell infiltrates [53]. The inducible CCM models, however, fail to model the localization of human CCM lesions; murine CCM lesions are primarily cerebellar or retinal [34,63,93,96].

To elucidate the mechanisms of CCM lesion development, inducible CCM mouse models have been crossed with reporter lines, such as *R26R-Confetti* mice. In the *R26R-Confetti* reporter line, recombined cells are labeled with one of four fluorescent proteins: nuclear GFP, cytoplasmic RFP, cytoplasmic YFP, and membrane-bound CFP [102,103]. Crossing inducible CCM3 mice with *R26R-Confetti* mice has revealed that CCM3-deficient endothelial cells undergo clonal expansion and form cavernomas [63,64]. Larger cavernomas are formed through CCM3-deficient endothelial cell-mediated recruitment of wild-type endothelial cells [63,64]. Use of reporter lines demonstrates that the second somatic mutation of *CCM3* leads to endothelial cell proliferation, resulting in the development of CCMs. 

Within the field, the most widely used model is the inducible CCM model. Using this particular CCM model to study CCM pathogenesis has resulted in a comprehensive understanding of cellular signaling pathways, such as MEKK3/KLF2/4 and PI3K/Akt/mTOR, contributing to CCM lesion formation [34,83]. Additionally, the use of a single CCM murine model across laboratories has increased the reproducibility of the findings. Basing CCM research on a single murine CCM model, however, neglects to capture the complexity of human CCM pathogenesis. Inducible CCM mice have proven useful for studying CCM lesion formation, as CCM lesions develop consistently and reliably [34,96]. While studying CCM lesion formation is important for understanding disease onset and pathogenesis, investigating mechanisms of lesion progression is essential, because leakage of CCM lesions results in patient symptoms and complications [73,74,75]. The early mortality and lack of stage 2 multicavernous lesions make the inducible CCM model unfit for studying lesion progression [34,93,96,100]. Other CCM models, such as the sensitized model, are advantageous for studying lesion progression and testing therapeutics, due to their longer lifespans and development of multicavernous stage 2 lesions [81,82,86,93,104]. Lesion formation, however, is difficult to study in sensitized CCM mice because of the stochastic manner in which the second hit and lesion formation occurs [93]. Harnessing the strengths of the sensitized model and the inducible model will provide a holistic understanding of CCM pathogenesis. 

### 5.2. Zebrafish Models

The ability to easily manipulate zebrafish genetics makes them a useful model for studying the function of CCM proteins [105]. This section will provide an overview of the zebrafish homologs of the *Ccm* genes, along with the models used to study each homolog.

#### 5.2.1. KRIT1/CCM1

The zebrafish homolog of *KRIT1*, *santa* (*san*), is a 741 amino acid protein with 16 coding exons [36]. Notable protein domains of san include two NPxY motifs, three ankyrin repeats, and a C-terminal B41/FERM domain [36]. Genetic screening for mutations affecting cardiovascular development led to the identification of *san*, as mutations in *san* produces zebrafish with enlarged hearts, thin myocardial walls, and absent endocardial cushions [36,106,107,108]. Studying the function of San is achieved through *san* mutants and morpholino targeting. Commonly used *san* mutants include m775 and ty219c, which lack the C-terminal portion of the San protein, and t26458, which lacks most functional domains of San [36,105,108]. Morpholinos targeting the donor splice site of exon 14 and the *san* start codon have been used to successfully recapitulate the big heart phenotype observed in *san* mutants [36,105]. The *san* mutant phenotype is endothelial cell autonomous: wild-type endothelial cells transplanted into *san* mutants develop normally [105]. This agrees with findings from murine CCM models, which demonstrate that CCM is an endothelial cell autonomous disease [63,64]. Use of transgenic driver lines has also been advantageous in elucidating *san* function. For example, endocardial-specific *san* expression via *TgBAC(nfatc1:GAL4FF)^mu286^* rescues the big heart phenotype; however, cerebellar central arteries remain dilated [108].

#### 5.2.2. CCM2

*Valentine* (*vtn*), the zebrafish homolog of *Ccm2*, is a 455 amino acid protein, with 10 coding exons and a PTB domain [36]. Similar to *san*, *vtn* was identified via genetic screening for mutations affecting cardiovascular development [107]. The *vtn* mutants also have the big heart phenotype, thin myocardial walls, and lack endocardial cushions [36,107]. Commonly used *vtn* mutants include m201, where the introduction of a premature stop codon results in an incomplete PTB domain, and *ccm2^hi296^*, which have an insertion in the first *vtn* intron [36,84,105]. Unlike the m201 mutant, *ccm2^hi296^* mutants have variable vascular and heart dilation, ranging from mild to severe [105]. Morpholinos targeting *vtn* splicing also yield zebrafish with the big heart phenotype and are used to study Vtn function [36,84]. Unlike *san*, exogenous *vtn* mRNA expression rescues the big heart phenotype in embryos injected with *vtn* morpholinos [36]. Most recently, CRISPR-Cas9 technology has been used to generate zebrafish *ccm2* mosaic mutants. Although half of the mosaic mutants die from cardiovascular defects, the remaining half survive to adulthood and develop brain vascular lesions comparable to human CCMs [109]. 

#### 5.2.3. PDCD10/CCM3 

Zebrafish have two *PDCD10* homologs: *ccm3a* and *ccm3b*, which encode proteins sharing an amino acid identity of 94% and 92% to human CCM3, respectively [49,110]. Morpholino-mediated depletion of *ccm3a/b* or only *ccm3b* results in death at 24–36 hours postfertilization (hpf) due to embryonic defects; however, embryos lacking *ccm3a* develop normally [36,49,105]. Using the same morpholinos at a lower concentration yields *ccm3a/b* morphants with cardiovascular defects, but unlike *san* and *vtn* morphants, *ccm3a/b* morphants have mispatterned cranial vessels [50,110]. To mimic a common human *CCM3* mutation resulting in exon 5 truncation, morpholinos targeting exon 3 splicing in *ccm3a/b* mRNA were used. Expression of truncated Ccm3a/b produced embryos with the big heart phenotype with blocked circulation despite normal vascular patterning [49]. Similar to *vtn*, expression of exogenous *ccm3a/b* mRNA rescues the morpholino-mediated big heart phenotype [49].

#### 5.2.4. Other Zebrafish Models

While most zebrafish modeling depends on depletion or mutation of the Ccm protein of interest, recently developed technologies have used fluorescently tagged *ccm* constructs to visualize subcellular localization of Ccm proteins in vivo [111]. Strengths and weaknesses of the various CCM models are summarized in Table 1. 

## 6. Developing CCM Therapeutics

Currently, therapeutic options for CCM treatment remain limited. Existing therapies include symptom management, surgical resection of symptomatic lesions, and gamma knife surgery [1,3,4,5]. Although recent studies demonstrate favorable outcomes post-surgical resection, regardless of lesion localization, less invasive options, such as pharmacological treatments, do not exist [112,113]. Therapeutic strategies in development focus on lesion stabilization, rather than lesion genesis, for two reasons: (1) mosaicism of the cerebrovasculature makes it difficult to predict lesion development, and (2) CCM lesion progression results in lesion leakage, giving rise to patient symptoms and complications [62,65,73,74,75]. As a comprehensive overview of therapeutic strategies was recently reviewed by Venugopal and Sumi, this section discusses the most prominent therapeutic strategies in development [114].

### 6.1. RhoA Kinase Inhibition

The rationale behind targeting RhoA kinase and ROCK activity originated from increased actin stress fiber formation following KRIT1 depletion in vitro, along with ROCK hyperactivity in the lesional endothelium of fCCM and sCCM patients [30,32,51]. Because of this observed increase in RhoA kinase and ROCK activity, various studies were conducted where CCM murine models were treated with Fasudil, a ROCK inhibitor [81,82,100,104]. Fasudil-treated sensitized CCM1 and CCM2 murine models have a significant decrease in the number and volume of CCM lesions, along with a decrease in hemorrhage, immune cell infiltrates, and endothelial cell proliferation [81,82]. Specifically, Fasudil treatment decreases the number of multicavernous stage 2 lesions, suggesting Fasudil prevents CCM lesion progression [82]. Interestingly, Fasudil treatment may be sex-dependent, as Fasudil-treated male CCM2 mice have a significant reduction in the number of stage 2 lesions; however, significance is lost when including female CCM2 mice [82]. While a promising treatment for fCCM1 and fCCM2, Fasudil treatment does not decrease lesion burden in chronic inducible CCM3 murine models [100]. Treatment with atorvastatin, a more potent RhoA kinase inhibitor, or BA-1049, a ROCK2-selective inhibitor, does decrease lesion burden in both sensitized CCM1 and CCM3 murine models [104,115]. Atorvastatin is currently being used in a phase I/IIa clinical trial (ClinicalTrials.gov Identifier NCT02603328); however, the fact that atorvastatin is an HMG-CoA reductase inhibitor and may introduce undesired side effects should be considered [116,117].

### 6.2. Propranolol 

Using the success of propranolol in treating severe infant hemangiomas as their rationale, case studies have demonstrated that propranolol treatment prevents CCM hemorrhages and even reduces the size of vascular malformations [118,119,120,121,122,123,124]. In CCM animal models, propranolol treatment (1) rescues aberrant development of *ccm2* mosaic mutant zebrafish via blocking β1-adrenergic receptors and (2) decreases the multicavernous lesion burden of CCM3 murine models [125,126]. While a retrospective study found no difference in risk of CCM-related hemorrhage in patients on β-blockers compared to untreated patients, a recent prospective study determined an association between β-blocker usage and a decreased risk for hemorrhage or focal neurological deficits [127,128]. A pilot study is currently being conducted using propranolol to treat fCCM patients (ClinicalTrials.gov Identifier NCT03589014) [127,129].

### 6.3. Other Therapeutics

Histological examination of human CCM lesions demonstrates immune cell infiltrates, including macrophages, B lymphocytes, and T lymphocytes [21,130]. B lymphocyte depletion in sensitized CCM3 murine models decreases the number and size of stage 2 multicavernous lesions but does not affect lesion genesis [78]. Ultimately, this posits immune response modulation as a potential therapeutic in preventing CCM lesion progression. 

Through a four-step screening process involving machine learning, Vitamin D_3_ and tempol, a superoxide scavenger, were identified as novel therapeutics for treating CCM. Treatment with either Vitamin D_3_ or tempol significantly reduced lesions in an endothelial-specific CCM2 murine model [131]. Treating chronic inducible CCM3 murine models with vitamin D_3_ or tempol, however, had no effect on lesion development, either individually or in combination with Fasudil [100]. Despite conflicting results across CCM models, a phase II clinical trial with tempol is currently being conducted (ClinicalTrials.gov Identifier NCT05085561). 

Recently, co-existing somatic gain-of-function mutations in *PIK3CA* and loss-of-function *CCM* mutations were identified within the same cells of human CCM lesions [83]. Because of this increase in phosphatidylinositol-3-kinase (PI3K)-mTOR signaling, rapamycin, an mTOR inhibitor, was used to treat acute inducible CCM1 mouse models, reducing lesion formation by 75% [83]. 

The majority of therapeutic studies use sensitized CCM murine models, as their longer survival time allows for testing treatment efficacy over the course of months [81,82,104]. Sensitized CCM murine models also recapitulate progression of human CCM lesions, as demonstrated by chronic hemorrhage, the development of stage 2 multicavernous lesions, and immune cell infiltrates [86,87,88]. Interestingly, many therapeutics are tested in CCM3, not CCM1 or CCM2, murine models. This is for two reasons: (1) Fasudil shows promise at treating fCCM1 and fCCM2, but not fCCM3, highlighting the need to develop fCCM3-specific treatments, and (2) the underlying assumption that if a therapeutic successfully treats fCCM3, the most aggressive form of fCCM, it is also promising in treating fCCM1 and fCCM2. This second reason, however, fails to understand the differences in molecular mechanisms underlying fCCM3 compared to fCCM1 and fCCM2. The interaction of CCM3 with the STRIPAK complex, along with the failure of Fasudil in treating CCM3, suggests a different molecular mechanism underlying fCCM3 disease progression [60,100]. Taking the complexity of disease progression into consideration will aid in the development of new therapeutics for all CCMs.

## 7. Conclusions

Identifying loss-of-function mutations in *KRIT1/CCM1*, *CCM2*, and *PDCD10/CCM3* has propelled CCM research forward. Studying the endogenous functions of the CCM proteins has revealed cellular processes and signaling pathways contributing to CCM pathogenesis. Discovery of the CCM proteins has also facilitated the generation of murine and zebrafish models, with each modeling a different aspect of CCM disease pathogenesis. These models, especially ones portraying lesion progression, are useful for discovering and testing therapeutics for CCM lesion stabilization (Figure 1). Through exploiting the strengths of each animal model, a fuller understanding of molecular mechanisms driving CCM lesion formation and progression will be achieved, allowing for the development of effective therapies.

## Figures and Tables

**Figure 1 ijms-23-05000-f001:**
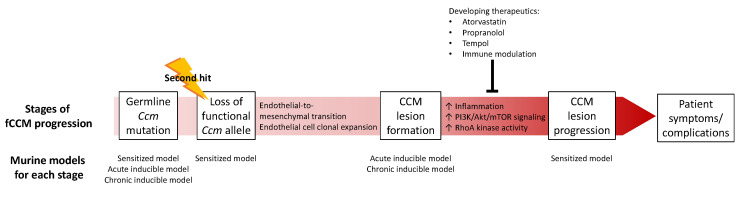
Schematic of CCM pathogenesis and the animal models best suited for studying processes of CCM disease.

**Table 1 ijms-23-05000-t001:** A summary of the types of CCM animal models, along with their strengths and weaknesses.

Species	Model Type	Modeling Strategy	Strength(s)	Weakness(es)
*Mus musculus* (Mouse)	Sensitized model	Crossing mice heterozygous for a *Ccm* allele with a genetically unstable mouse line (e.g., *Trp53*^−/−^ or *Msh2*^−/−^ mice)	Lesions localized throughout brain Models second-hit mechanism Lesions are comparable to human lesions histologically Models lesion progression	Difficult breeding scheme Low yield of animals with desired genotype Genetically unstable background can lead to tumor development Difficult to study lesion genesis
	Acute inducible model	Crossing *Ccm*-flox mice with CreERT2 lines; high TMX dose immediately post-birth	Well-defined timepoint of lesion onset good for modeling lesion genesis Lesions are comparable to human lesions histologically	Lesions restricted to cerebellum and retina Mice survive to P30 Difficult to study lesion progression
	Chronic inducible model	Crossing *Ccm*-flox mice with CreERT2 lines; low TMX dose post-birth	Survive to adulthood Lesions localized throughout brain Lesions are comparable to human lesions histologically	Primarily stage 1 lesions Few lesions with immune cell infiltrates Difficult to study lesion progression
*Danio rerio* (Zebrafish)	*ccm* mutants	*ccm* mutations identified via genetic screening	Easy genetic manipulation Rescue phenotype via exogenous mRNA	Defective cardiovascular development instead of brain vascular lesion development
	*ccm* morphants	Morpholino targeting to either deplete expression or mimic mutation	Easy genetic manipulation Rescue phenotype via exogenous mRNA Targeting splice sites can mimic human mutations	Defective cardiovascular development instead of brain vascular lesion development
	*ccm* mosaic mutants	CRISPR-Cas9-mediated *ccm* deletion	Develop brain vascular lesions Mimic mosaicism of human CCM lesions	Low survival rate (~50% die from cardiovascular defects)

## Data Availability

Not applicable.

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
