# Peer review of "Cerebral Cavernous Malformation Pathogenesis: Investigating Lesion Formation and Progression with Animal Models"

_ijms, 2022, doi:10.3390/ijms23095000_

Round 1

Reviewer 1 Report

The article is devoted to investigation of cerebral cavernous malformation (CCM), a cerebromicrovascular disease that affects up 13 to 0.5% of the population. In this article, the authors consider various forms of this disease, analyze the consequences of mutations in key genes that regulate the development of this disease. This review provides an overview of CCM disease pathology and CCM protein function, along with a detailed discussion of murine and zebrafish CCM models and promising therapeutics to treat CCM. 
The article is interesting, well written. 
A serious note for the article is that more than 80% of the references is not included in the 5-year span 2017-2021. The topic of this article is actual, does not refer to does not assume outdated information. Over the past 5 years, there may be new data on this issue, with which you need to compare the results of this study. I am sure that it is necessary to review the current literature and add the latest data to this article.

Author Response

We thank the reviewers for their critiques and suggestions (in bold below). Our responses to each point are in plain type. In the main text, changes are tracked and highlighted in red.

The article is devoted to investigation of cerebral cavernous malformation (CCM), a cerebromicrovascular disease that affects up 13 to 0.5% of the population. In this article, the authors consider various forms of this disease, analyze the consequences of mutations in key genes that regulate the development of this disease. This review provides an overview of CCM disease pathology and CCM protein function, along with a detailed discussion of murine and zebrafish CCM models and promising therapeutics to treat CCM. The article is interesting, well written. 

Thank you very much for your positive comments.

A serious note for the article is that more than 80% of the references is not included in the 5-year span 2017-2021. The topic of this article is actual, does not refer to does not assume outdated information. Over the past 5 years, there may be new data on this issue, with which you need to compare the results of this study. I am sure that it is necessary to review the current literature and add the latest data to this article.

Thank you for your comment. We updated our references to include studies from 2017-2022. Please see highlighted in red reference numbers: 18, 31,33,73,80, 81, 96, 98-100, 102, 113-116, 125, 127, 129.

Reviewer 2 Report

This is a very interesting and up-to-date review of the current knowledge of CCM disease and related proteins, also based on the evaluation of the different animal models used to study CCM pathogenesis, from a translational point of view about possible therapeutic targets. I think that the manuscript could provide some insight into the preclinical model, also for physicians involved in the management of cavernomas.

I only suggest adding the citation of a very recent review  (Venugopal V. et al. Molecular Biomarkers and Drug Targets in Brain Arteriovenous and Cavernous Malformations: Where Are We? Stroke. 2022 Jan;53(1):279-289. doi: 10.1161/STROKEAHA.121.035654), to highlight some new information about the basis of possible therapeutic management

Author Response

We thank the reviewers for their critiques and suggestions (in bold below). Our responses to each point are in plain type. In the main text, changes are tracked and highlighted in red.

This is a very interesting and up-to-date review of the current knowledge of CCM disease and related proteins, also based on the evaluation of the different animal models used to study CCM pathogenesis, from a translational point of view about possible therapeutic targets. I think that the manuscript could provide some insight into the preclinical model, also for physicians involved in the management of cavernomas.

Thank you very much for your positive comments.

I only suggest adding the citation of a very recent review  (Venugopal V. et al. Molecular Biomarkers and Drug Targets in Brain Arteriovenous and Cavernous Malformations: Where Are We? Stroke. 2022 Jan;53(1):279-289. doi: 10.1161/STROKEAHA.121.035654), to highlight some new information about the basis of possible therapeutic management

We included suggested reference (number 115).